# From little girls to adult women: Changes in age at marriage in Scheduled Castes from Madhya Pradesh and Uttar Pradesh, India

Grażyna Liczbińska[1]*, Marek Brabec[2,3], Rajesh K. Gautam[4], Jyoti Jhariya[4], Arun Kumar[4,5]

1 Institute of Human Biology and Evolution, Faculty of Biology, Adam Mickiewicz University, Poznań, Poland, 2 Department of Statistical Modelling, Institute of Computer Science, The Czech Academy of Sciences, Prague, Czech Republic, 3 Department of Biostatistics, National Institute of Public Health, Prague, Czech Republic, 4 Department of Anthropology, Dr. Harisingh Gour Vishwavidyalaya (A Central University), Sagar, Madhya Pradesh, India, 5 Sports Authority of India, National Centre of Excellence, Bhopal, MP, India

* grazyna@amu.edu.pl

**Data Availability Statement:** Data cannot be shared publicly because of many confidential data and potentially sensitive information. Data generated in this study are available by contacting

## Abstract

### Background

Research confirms the negative relationship between early marriage and mothers' and children's health outcomes. This is why studies of the changes in age at marriage are an important task from the point of view of the health status and well-being of a mother and her offspring, especially in groups represented by extremely disadvantaged social strata in India. The results of such studies may influence the future family planning policy in the country.

### Objectives

This study aims to investigate the trend of age at marriage among the Scheduled Castes (SCs) women from two Indian states: Madhya Pradesh and Uttar Pradesh relative to the level of education and also to socioeconomic changes in the states. These states manifest the highest proportion of girls getting married below the age of 18 years–far above the proportion observed in entire India.

### Methods

Women from Scheduled Caste, N = 1,612, aged 25–65, born in 1950–1990 were investigated. A modern semiparametric regression approach was used. To capture the relationship between age at marriage and year of birth, categories of women's level of education (illiterate; primary: 1st–5th standards; middle school: 6th–8th standards; high school: 9th–10th standards; higher secondary: 11th–12th standards), and categories of the profession (women working in the agricultural sector or the non-agricultural sector), flexible framework of the Generalized Additive Model (GAM) was applied.

### Results

A significant impact of the cohort defined by the year of birth (<0.001), and women's education (<0.001) on age at marriage was noted, while the influence of women's occupation was

the first author, Grażyna Liczbińska, if requested reasonably (contact via grazyna@amu.edu.pl). All the data are also deposited to the Research Committee of the Department of Anthropology, Dr. Harisingh Gour Vishwavidyalaya (A Central University), Sagar (MP), India; which can be accessed by requesting the Head of the Department or the Chairman of the committee (email: hodanthropology@dhsgsu.edu.in).

**Funding:** Research reported in this publication was supported by The Ministry of Human Resources of the Government of India, Delhi under Award Number No. 174046N04 (PI: Rajesh Kumar Gautam; co-PI: Grażyna Liczbińska), by the Rajiv Gandhi National Fellowship under Award Number No.F.16–29/2006 (SA–II)/817) (PI: Jyoti Jhariya), by the Post-doctoral Fellowship for Women (No. F. No.15–2/2012) (PI: Jyoti Jhariya), and by the Institute of Computer Science Czech Republic under Award Number RVO 67985807 (PI: Marek Brabec). The funders had no role in study design, data collection and analysis, decision to publish, or preparation of the manuscript. There was no additional external funding received for this study.

**Competing interests:** The authors have declared that no competing interests exist.

not significant (p = 0.642). Mean age at marriage differed significantly with different education level. Women who graduated from primary school married 0.631 years later on average than illiterate ones, while those who graduated from middle schools, high schools (9th–10th standards) and higher secondary schools married significantly later than illiterate ones by 1.454 years and 2.463 years, respectively. Age at marriage increased over time: from slightly above 15 years in the cohort of illiterate women born in 1950 to almost 19 years in quite well-educated women born in 1990. The average age at marriage estimated for four education levels in 1990 ranged between 16.39 years (95%CI: 15.29–17.50) in the group of illiterate women and 18.86 years (95%CI: 17.76–19.95) in women graduated from high and higher secondary schools.

## Conclusion

The rise of age at marriage can be partly explained by the increase of females enrolled in schools, the alleviation of poverty, and the implementation of social programs for women.

## Introduction

The mating system and matrimony are strongly influenced by cultural modifiers [1–8]. Early marriage of women (<18 years) is a fairly common practice in many developing countries [9–11]. Research confirms the negative correlation between early marriage and both mother's and child's health outcomes [9, 12–18] and children's schooling outcomes [9, 19, 20]. The link between early marriage and high domestic violence is also emphasized in studies [21–24]. Early age at marriage reduces mothers' educational attainment, which can adversely affect the educational outcome of their children [20]. Raj and a team [25] found a higher risk of malnutrition in children born to mothers married very early compared to those born to women married at maturity. Research on Pakistani girls has shown that young matrimony increases the morbidity and mortality of their children [26]. Evidence shows that early marriage is a substantial barrier to the social and economic development in many countries and important for woman's and child's social and health outcomes. According to *A Statistical Analysis of Child Marriage in India* [27: 1–2], "marriage at a young age has far-reaching consequences for both girls and boys in terms of their overall development and in making important life decisions and securing basic freedoms, including pursuing educational opportunities, earning a sustainable livelihood and accessing sexual health and rights. Child marriage has a profound physical, intellectual, psychological, and emotional impact on both boys and girls. It results in cutting off childhood, educational opportunities, and attainment of employment and curtails the role the children can take as adults to productively contribute to their development and the economy of the country. For girls, it enlarges their fertility span, which almost certainly results in premature pregnancy and multiple pregnancies and is likely to lead to a lifetime of domestic and sexual subservience over which they have no control. Combined with malnutrition, it often results in physical wastage, birth complications and delivering of low birth-weight babies with reduced chances of survival, adding to both higher infant mortality and maternal mortality ratio."

Since age at marriage is significantly related to the level of fertility and life chances, encompassing for instance access to education, the topic concerning the moment of entry into matrimony is of interest to anthropologists, human biologists, and sociologists [28]. This study aims

to investigate the trend of age at marriage among the Scheduled Castes (SCs) women from two Indian states: Madhya Pradesh and Uttar Pradesh relative to the level of education and also to socioeconomic changes in the states. We selected these particular states as they manifest the highest proportion of girls marrying below the age of 18 years–far above the proportion observed in entire India [29]. Research on the changes in age at marriage among the lowest social groups in India is important from the point of view of health status and well-being of a mother and her offspring in the extremely disadvantaged strata. The results of such studies may influence the future family planning policy in the country.

## Materials and methods

### Study design and participants

The research was conducted in 2007, 2015, and 2017 under the auspices of the Department of Anthropology at Dr. Harisingh Gour Vishwavidyalaya University (A Central University), Sagar, Madhya Pradesh. It was based on empirical fieldwork. Various anthropological methods were used for the collection of information on females: observation, participation, semi-participation, case study, and group discussion [3]. Data were obtained for women from the state of Madhya Pradesh located in central India and Uttar Pradesh located in north India. Both are economically and sociologically backward, with the highest proportion of girls marrying below the age of 18 years. Moreover, the governments of Madhya Pradesh and Uttar Pradesh have not taken any initiative to make the registration of marriages compulsory [29]. All females belong to Scheduled Castes, one of the most disadvantaged socio-economic groups in India [30].

Primary data on various demographic characteristics were collected using a semi-structured schedule that was culturally validated after pilot surveys. Different projects had different objectives, hence the tools also varied, but the part containing demographic information was similar in all three schedules. Similar information on age at marriage was collected by three different investigators. For three projects, the data were collected by door to door survey, whereas for one of the projects the data were collected in the University campus. Multistage stratified sampling method was adopted in the selection of the area and population. The villages were selected by PPS (proportion to population size) sampling method. Further, within a village, the purposive sampling method was adopted for the recruitment of the respondents; since some of the respondents rejected the offer to participate in the study, they were replaced by identical respondents who were willing to participate. Some of the respondents were excluded as they did not meet the criterion of ideal respondents as per objectives of the study. All projects were conducted under the supervision of a senior faculty member and followed the appropriate ethical considerations. For projects 1–3, the data were collected in the district of Mandla and Sagar, Madhya Pradesh, for project 4, the data were collected in the district of Banda, Uttar Pradesh state of the Indian Union.

All precautions were taken to minimize biases, still, some inter-investigator bias cannot be ruled out completely. In Indian society, marriage is an important event, but there is no obligatory system of marriage registration. Hence, the information provided by the respondent was the only authentic source of information, which was further validated using the genealogical method of age estimation: age of elderly people was traced from the known age of younger individuals. A genealogy or a family tree is drawn for 3–4 generations. Usually, the age of individuals of younger generation is accurately known. To determine the age of elder individuals the reference is taken from the known age, e.g., if an illiterate mother's accurate age is doubtful, it can be inferred from the age of her child. Suppose a child is 10 years old and he/she is her first child. Mother's age will be between 25–30 years (can't exceed 30 years). A female can't

give birth before puberty. She can't give birth just after puberty or at the age of the first menarche either. So, the earliest age at first birth cannot be earlier than 15 years; 16 to 17 years if the menarche is delayed. In such circumstance, the birth can also be delayed, but not more than until the age of 20 years. Hence, current maternal age must be in between 25–30 years. This error margin of 5 years can be further reduced by obtaining the average age of mothers at the time of the birth of the first child. If she gave birth at the age of 17 years, at the time of survey her accurate age must have been 27 years. In this way the accurate age of individuals is obtained by using the genealogy method.

According to the nationwide survey (NFHS-3) [31], the majority of women marry before the age of 45 years (99%). Around 1% of women remain unmarried for various reasons. As marriage is an important event most women remember its occurrence, which was confirmed by their evidence. The age at marriage was recorded in years. If the age at marriage was 15 years and 6 months or less it was rounded down to 15 years, whereas if the age at marriage was 15 years and 7 months or more, it was rounded up to 16 years.

A total of 84.7% of females surveyed provided information on their age at marriage; the remaining either did not provide any information or the information provided was found questionable during crosscheck and hence removed from the analysis. After the completion of the data collection and data entry some information from respondents was found dubious, including age at marriage, hence respondents were visited again to crosscheck the information. If respondents were not found in their residence or the information they provided was still uncertain, such individuals were removed from the analysis. Therefore, the analysis was limited to 1,612 females aged 25–65, born in 1950–1990. Apart from age at marriage, information on women's level of education and occupation was recorded. The level of education was divided into four categories: (1) illiterate, (2) primary (1st–5th standards), (3) middle school (6th–8th standards), (4) high school (9th–10th standards), and higher secondary (11th –12th standards). In India societal attitudes (especially in lower social strata) still prioritize early marriages over work and education for women [32]. This is why women's level of education is considered in this work a strong proxy for a woman's social position and a determinant of certain social changes in India, particularly in groups located at the very bottom of the stratified Indian society [33, 34]. Women's occupation was summarized in two categories: (1) women working in the agricultural sector (e.g., agriculture laborers, farmers), (2) women working in the non-agricultural sector (e.g., self-employed, small-scale vendors, private job holders).

## Statement of ethics

All procedures contributing to this work comply with the ethical standards of the relevant national and institutional committees on human experimentation and with the Helsinki Declaration of 1975, as revised in 2008. The study was approved by the respective Departmental Research Committee (DRC)/ Thesis Advisory Committee (TAC)/ Research Advisory Committee (RAC) of the Department of Anthropology, Dr. Harisingh Gour Vishwavidyalaya (A Central University), Sagar, MP, India. The data were collected as per ethical guidelines. The participants were interviewed at their residences in their local language and dialect (i.e., Hindi and Bundeli), with ease for a respondent given priority. The original English version of the questionnaire was translated in Hindi and Bundeli by two bilingual individuals. The translated questionnaires were then back-translated by two other bilingual experts into English. This procedure was repeated until the back-translations agreed with the original version. The informed written consent was obtained from each of the participants. Simultaneously, written consent was also obtained from the village head and administrative authorities at the district level.

## Characterization of Scheduled Castes

The Indian sub-continent is dominated by pan-Hinduism, and Casteism is an inherent feature of Hinduism [35]. It is a stigma in Indian society and a great hindrance to its development, as the society is highly stratified and divided based on castes. The ancient Hindu society initially distinguished four (later five) mutually exclusive, hereditary, endogamous, and job specific Varnas, which in English means castes: Brahmins (priests, teachers), Kshatriyas (warriors, royalty), Vaisyas (moneylenders, traders), Sudras (menial jobs), and Ati Sudras (the untouchable, doing the lowest of menial jobs) [36]. "Caste affiliation dictated all aspects of a person's existence. The Varna hierarchy was relatively straightforward, with the first three tiers considered superior to the last two. This organization corresponds to a very rudimentary economy" [37: 322].

In 1950 the Indian Constitution listed 1,108 groups belonging to Scheduled Castes (formerly untouchable, or lowest castes) in 25 states [38]. According to the census of 1991, 16–17% of the Indian population was classified as Scheduled Castes. In the state of Madhya Pradesh, 48 ethnic groups belong to Scheduled Castes, which constitutes around 15.6% of the state population [39]. In the state of Uttar Pradesh, 66 groups are categorized as Scheduled Castes, which makes up 20.6% of the population of Uttar Pradesh [39]. Ethnic groups such as Balai, Chamar (Jat), Dhobi, Domar, Khati, Koli, Kumbhar, Kori, Mahar, Mehra, and Satnami are categorized as Scheduled Castes [33, 35, 40, 41].

The Scheduled Castes are one of the most disadvantaged socioeconomic groups in India [30]. They are placed bottom of the social strata, after Brahmins (placed at the top), Kshatriyas, and Vaishyas (placed in the middle). They live in very poor conditions suffering several health and nutritional problems [41]. Many are still regarded as untouchable/unclean. Although untouchability has been abolished by law [42] it still exists among Hindus, particularly in rural areas [40]. The main livelihood of Scheduled Castes involves agriculture, though many also work as day laborers. Nevertheless, some Schedule Castes members practice traditional professions, e.g., Chamars–shoe making and repair, Kumbhars–pottery, and Koris–weaving, and Dhobi–washing clothes, known sometimes as "washer men" [41].

## Statistical analysis

For the analysis of the empirical data, we used modern semiparametric regression approach. The relationship of marriage age to the various explanatory variables is generally unknown, and possibly nonlinear. To capture it without systematic distortion (which might easily occur when modelling relies on a priori knowledge or pre-supposed functional form like simple linearity), we used flexible framework of the Generalized Additive Model (GAM) [43, 44]. The nonparametric part was applied via the roughness-penalized spline approach [45, 46]. The model was fitted by optimizing penalized likelihood [44]. Unknown penalty coefficients were estimated via generalized cross-validation [47]. The R statistical environment [48] and the *mgcv* library [44] were used for modelling and data preparation.

## Results

Frequencies of women marrying at age $< 18$ years vs $\geq 18$ years by their level of education were as follows: illiterate, 71.31% vs 28.69%; primary (1st–5th standards), 44.85% vs 55.15%; middle school (6th–8th standards), 35.04% vs 64.96%; high school (9th–10th standards), and higher secondary (11th–12th standards), 23.66% vs 76.34% ($\chi^2$ = 214.92, df = 3, p<0.0001).

We observed statistically significant impact of the year of birth (<0.001) and women's education (<0.001) on average age at marriage, while the influence of women's occupation was

**Table 1. Parametric coefficients for women's SES effects on age at marriage.**

| Effect name | N | Estimate | Standard error | p-value |
|---|---|---|---|---|
| **Intercept** | 1,612 | **15.826** | **0.110** | **<0.000**[***] |
| Education 2 | 1,612 | 0.631 | 0.238 | 0.007** |
| Education 3 | 1,612 | 1.454 | 0.328 | 0.000*** |
| Education 4 | 1,612 | 2.463 | 0.451 | 0.000*** |
| Occupation 2 | 1,612 | -0.132 | 0.283 | 0.642 |

**, *** statistically significant values. Education: 2 –primary (1st–5th standards), 3 –middle school (6th–8th standards), 4 –high school (9th–10th standards), and higher secondary (11th–12th standards); Occupation: 2 –women working in the non-agricultural sector.

not significant (p = 0.6419). The effect of women's socioeconomic status (SES; education and profession) is presented in Table 1.

Women who graduated from primary school (1st–5th standards) married 0.631 years later than illiterate ones. Similarly, women who graduated from middle schools (6th–8th standards), high schools (9th–10th standards), and higher secondary schools (11th–12th standards) married significantly later than illiterate ones by 1.454 years and 2.463 years on average, respectively. The age at marriage increased over time (Fig 1).

As shown by an additive model, illiterate women married the earliest, while the latest were those graduated with high school and higher secondary education (Fig 2). The mean age at marriage of illiterate women increased from slightly above 15 years in the cohort born in 1950 to above 16 years in women born in 1990. In the group who graduated from high school and higher secondary, the age at marriage increased from above 17 years in 1950 to almost 19 years in 1990.

The average ages at marriage estimated for four education levels in 1990 are presented in Table 2 (Fig 3). For example, for illiterate women, this was 16.39 years (95%CI: 15.29–17.50), while for women graduated from high schools and higher secondary schools– 18.86 years (95%CI: 17.76–19.95).

## Discussion

Early marriage refers to a situation when a child younger than 18 enters into matrimony. Child matrimony in India has been practiced for centuries and involves complex religious traditions, social practices, economic factors, and deeply rooted prejudices. According to scholars, the practice of child marriages began only after 600 AD under the influence of the writings of *Dharmasutras* and *Smiritis*. From the 7th century onwards, child marriage came under the influence of religious scripture spread in Indian society. The virginity of the girl during marriage was considered above her health and education. It was the prime job of the parents to marry off their daughter before she attained puberty or menstruation. The practice continued unabated through the Middle Ages to the British period. In the latter period, child marriages experienced further degeneration leading to the practice of infant marriages. Child marriages were eventually banned in 1894 [28].

In India, many girls marry before they attain physical and mental maturity and frequently experience domestic violence [10]. According to statistics, India has the largest population of child wives in the world [49]. Early marriage is a phenomenon observed mainly among the poorest social groups [10, 25]. According to Devi and Singh [10], poverty is the main reason parents chose to find a spouse for their daughters. Matrimony is associated with the reduction of expenses for an additional family member, including expenses for the education of girls [10].

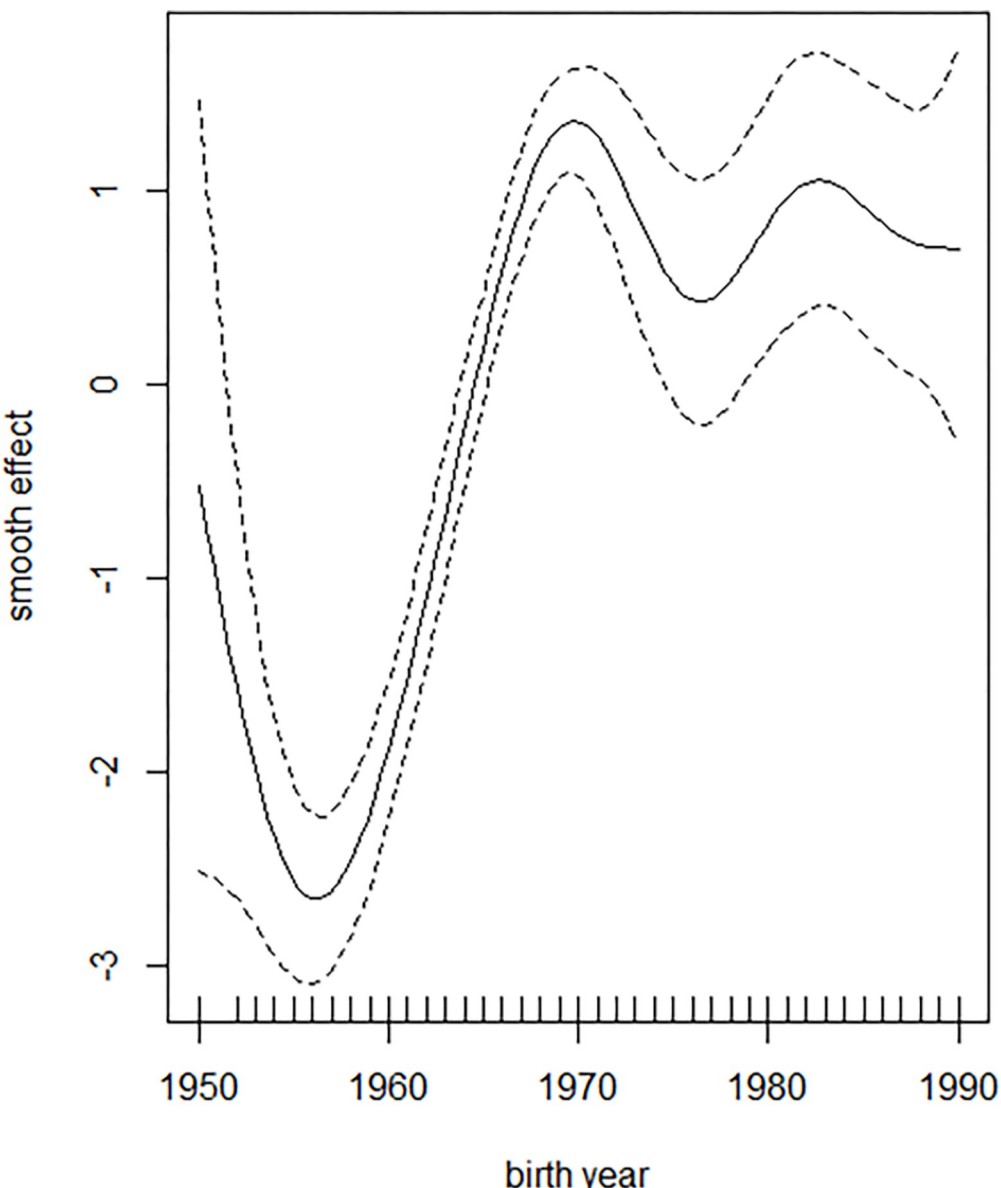

**Fig 1. Smoothed estimated effect of the birth year on age at marriage.** The solid line is an estimate of the effect of birth year on age at marriage, while the dotted lines indicate the 95% confidence intervals.

There are several reasons for the increased age at marriage among females of Scheduled Castes. Firstly, it occurred due to changes in legal regulations regarding age at marriage and the restrictions enforced to prevent child marriages. Secondly, it was a result of women's education programs being launched to promote the education of girls, which increases their chance of working outside the agricultural sector. Thirdly, the economic situation of many social segments in India, including Scheduled Castes, improved [29].

## Changes in legal regulations

The United Nations took up the matter of regulating age at marriage already in the 1960s. The 1962 *Convention on Consent to Marriage, Minimum Age for Marriage and Registration of*

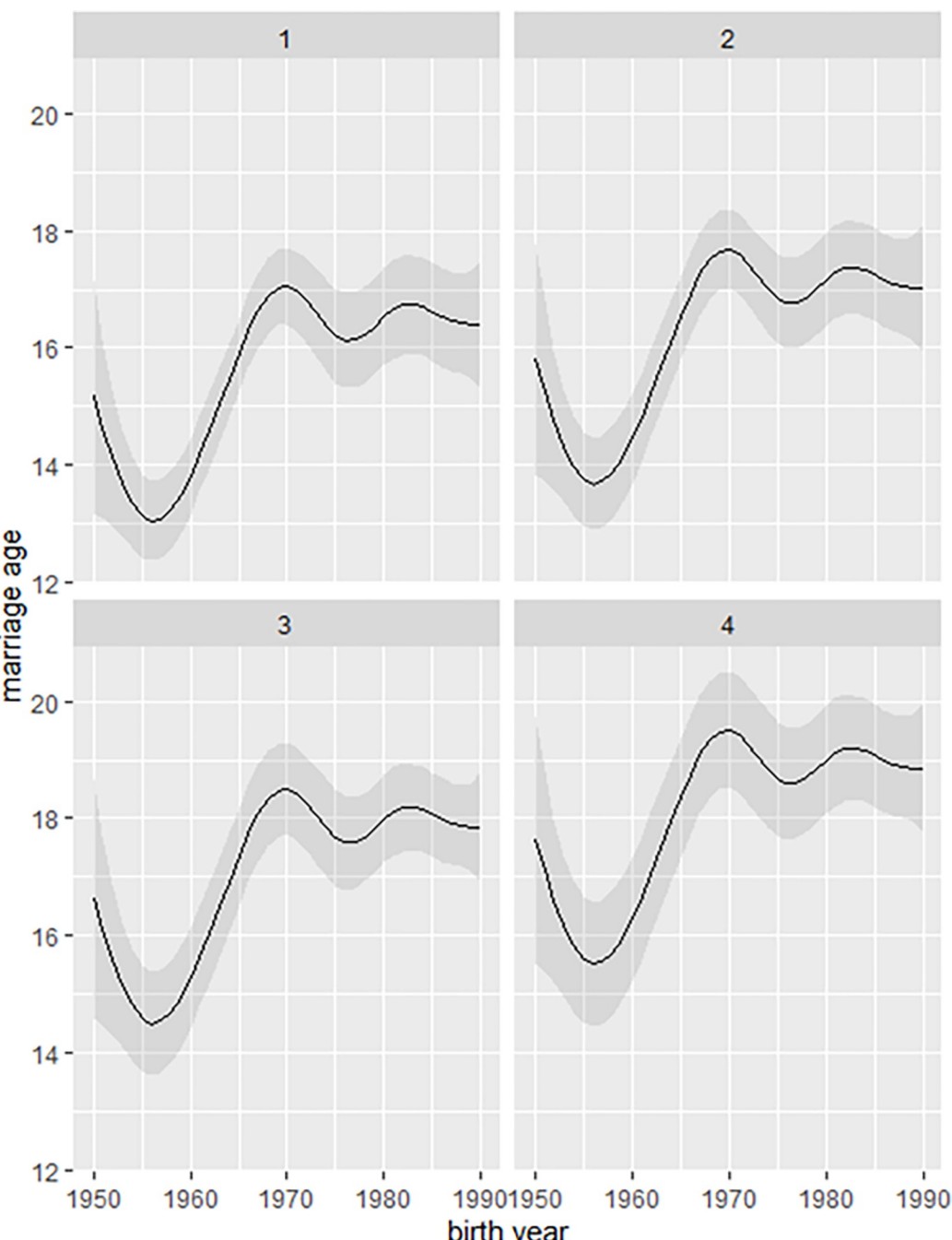

**Fig 2. Changes in average ages at marriage according to women's level of education.** The solid lines are the estimated ages at marriage, while the shaded lines indicate the 95% confidence intervals. 1–4 represent women's level of education: 1– illiterate, 2 –primary (1st–5th standards), 3 –middle school (6th–8th standards), 4 –high school (9th–10th standards), and higher secondary (11th–12th standards).

*Marriages* states that: "(1) Men and women of full age, without any limitation due to race, nationality, or religion, have the right to marry and to establish a family. They are entitled to equal rights as to entering into marriage, during marriage, and at its dissolution;

(2) Marriage shall be entered into only with the free and full consent of the intending spouses (. . .)" [50: 232]. The *Convention* came into force in December 1964 and obliged states

**Table 2. Estimates of the average ages at marriage in 1990 for different education levels.**

| Education level | N | Average age at marriage (evaluated in 1990) | Lower limit of 95% CI | Upper limit of 95% CI |
|---|---|---|---|---|
| 1 | 1,612 | 16.39 | 15.29 | 17.50 |
| 2 | 1,612 | 17.03 | 15.95 | 18.11 |
| 3 | 1,612 | 17.85 | 16.90 | 18.80 |
| 4 | 1,612 | 18.86 | 17.76 | 19.95 |

1– illiterate, 2 –primary (1st–5th standards), 3 –middle school (6th–8th standards), 4 –high school (9th–10th standards), and higher secondary (11th–12th standards).

to specify a minimum age at marriage, to prohibit legal acceptance of any marriage without the full and free consent of both parties, and to register all marriages contracted [29]. In India the Child Marriage Restraint Act (CMRA; known as the "Sarda" Act, 1929), originally prohibited the marriage of girls and boys below15–18 years of age, respectively. In 1978, the age at marriage for girls increased to 18 years and for boys to 21 years [10]. Among the contraindications of entering into marriage very early are early pregnancy, health complications, inability to plan a family, the high difference in age between a bride and bridegroom, and the impact on the health of women and girls vulnerable to HIV infections [10]. The Prohibition of Child Marriage Act, 2006, which is the national law against child marriage, does not allow asking minors for consent and treats every child marriage as a punishable offense [29].

## Women's education

In our research, the age of entering matrimony in the Scheduled Castes increased over time and significantly differed according to women's level of education. Education is an important factor responsible for the increment in the age at marriage. Along with colonial rule in India, the path of education was opened to the Scheduled Castes and women. After independence in 1947 women's access to education further increased, availing the opportunity for women from deprived social strata like SCs to be educated. To reduce childhood marriages or marriages at an early age, the state governments promoted the education of girls by offering different incentives to the mother, family, and the girl herself. This led to an increase in the rate of literacy. Changes in the literacy rates over time in Madhya Pradesh and Uttar Pradesh are presented in Table 3. In six decades (1950s–2010s) they increased more than five times in both states [51]. In 2011, female literacy rates in Madhya Pradesh and Uttar Pradesh were at the level of 59.2% and 57.2%, respectively [52].

In entire India the number of girls aged 6–10 years enrolled in primary schools (classes I–V) increased twelve times in 2012 compared to 1950. At the level of upper primary education (classes VI–VIII) the number of girls attending school at the age of 11–13 years increased in 2012 by sixty times compared to 1950 [53] (Table 4).

Since the 1970s there has been a significant improvement in education in SCs [54]. However, there is still a large disparity between the number of girls and boys attending school. In entire India, in 2011–2012 the number of females from Scheduled Castes enrolled in primary schools in grades I–V was almost four times higher than 1980–1981. The number of SCs girls enrolled in upper primary schools (grades VI–VIII) increased ten times for the same period, while at the level of senior secondary education, i.e., girls aged 14–15 (grades XI–XII) between 2000/01 and 2011/12 increased almost four times [53] (Table 4).

Education and awareness have played a pivotal role in discouraging child matrimony. Nationwide awareness-raising programs regarding child marriage, the so-called *Bal Vivah Virodh Abhiyan* (English: *Campaign against Child Marriage*), run by the National Commission

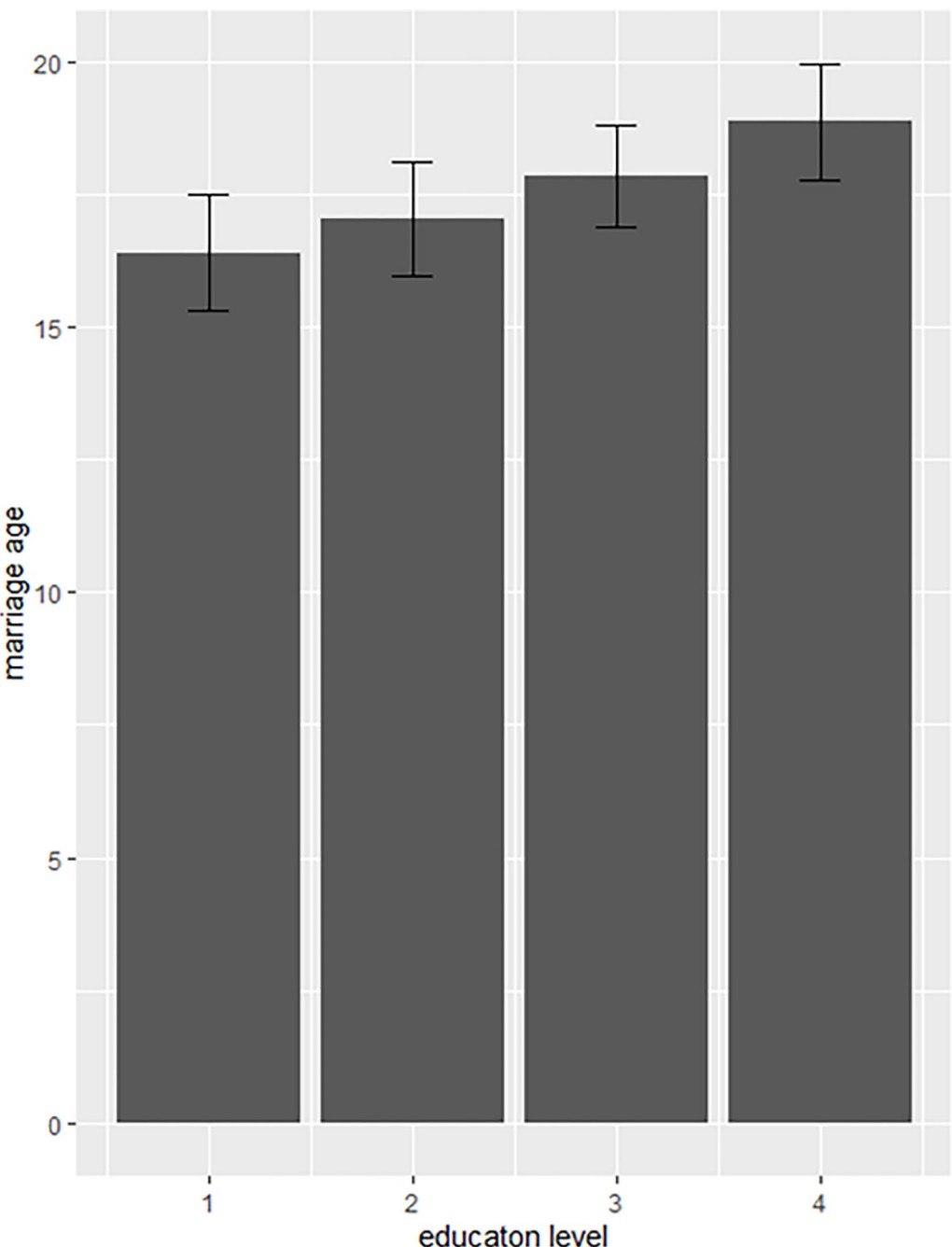

**Fig 3. Average age at marriage by education level.** The upper and lower hinges represent the upper/ lower limits of 95% CI. 1–4 represent women's level of education: 1– illiterate, 2 –primary (1st–5th standards), 3 –middle school (6th–8th standards), 4 –high school (9th–10th standards), and higher secondary (11th–12th standards).

for Women and focussed particularly on girls and women, play a very important role among parents. An increasing number of different programs are being launched and implemented for adolescent girls, such as for example, the *Rajiv Gandhi* Scheme for the Empowerment of Adolescent Girls–"*Sabla*", "*Kishori Shakti Yojana*", "*Dhanalakshmi*", "*Betipadayobetibadayo*", etc [29].

**Table 3. Changes in literacy rates (%) in Madhya Pradesh and Uttar Pradesh.**

| Year | Madhya Pradesh | Uttar Pradesh |
|------|----------------|---------------|
| 1951 | 13.16 | 12.02 |
| 1961 | 21.41 | 20.84 |
| 1971 | 27.27 | 23.99 |
| 1981 | 38.63 | 32.65 |
| 1991 | 44.67 | 40.71 |
| 2001 | 63.74 | 56.27 |
| 2011 | 69.32 | 67.68 |

Source: [51: 8].

## Decline of poverty

Widespread poverty is also one of the main factors responsible for early marriage in India. According to the Centre for Child Rights statistics, the top five states with the highest GDP ranking have a lower rate of child marriage. Uttar Pradesh and Madhya Pradesh belong to Indian states with higher poverty and a higher than national average proportion of matrimonies of girls below the age of 18 years, i.e., 33% and 29%, respectively which means that more than one-fourth of the girls in these states are married before the age of 18 years [29]. Apart from poverty, the practice of "Dowry" also contributes to a higher number of child matrimonies [29]. The dowry system in India refers to property that the bride's family gives the groom, his parents and relatives as a condition of marriage [55, 56]. The gradual reduction of poverty, however, has resulted in a decline in child marriages. In 1973–1974 the poverty ratio declined from 56.4% and 49% in rural and urban areas, respectively to 27.1% and 23.6% in 1999–2000. At the start of the 1970s almost 62% of the population of Madhya Pradesh and 57% of that in Uttar Pradesh lived below the national standard. By the late 20th and early 21st centuries, the poverty ratio in these two states was 37% and 31%, respectively [57]. In the rural areas of Madhya Pradesh consumption expenditures increased from 50 INR (Indian Rupee) per person per month in 1973–1974 to 401.50 INR per person per month in 1999–2000, while in the rural areas of Uttar Pradesh, the increase was from 51.3 INR per person per month in 1973–1974 to

**Table 4. Changes in the numbers of females enrolled in education in India.**

| | 1950–1951 | 1980–1981 | 2000–2001 | 2010–2011 | 2011–2012 |
|---|---|---|---|---|---|
| **All students** | | | | | |
| Classes I–V | 54 | 285 | 498 | 646 | 672 |
| Classes VI–VIII | 5 | 68 | 175 | 292 | 299 |
| Classes IX–X | NA | NA | 74 | 143 | 155 |
| Classes XI–XII | 2 | 34 | 38 | 86 | 94 |
| Higher education | 0 | 13 | 32 | 120 | 130 |
| **Scheduled Castes** | | | | | |
| Classes I–V | NA | 38 | 91 | 129 | 139 |
| Classes VI–VIII | NA | 6 | 26 | 53 | 59 |
| Classes IX–X | NA | NA | 11 | 26 | 31 |
| Classes XI–XII | NA | NA | 5 | 13 | 18 |
| Higher education | NA | NA | NA | 13 | 16 |

Source: [53: 23–25]. Note: data in columns are presented in lakh; 1 lakh = 100,000.

466.6 INR per person per month in 1999–2000 [57]. The same increase was noted in the urban areas of Madhya Pradesh and Uttar Pradesh: from 65.5 INR and 60.8 INR per person/month to almost 695 INR and 690 INR/person/month, respectively [57]. In 1983 as many as 59% of members of SCs were living in poverty; in 1993–1994 poverty noted in 48.6% of SCs, while in 2004–2005 poverty had declined to 37.1% in SCs [57]. Infant death rates are considered an excellent indicator of living conditions: in rural and urban areas in 1972 there were 150 and 85 deaths per 1,000 live births, respectively, but by the end of 1999 these values had been reduced by half [57]. In addition, in India life expectancy at birth increased from slightly above 40 years in the 1970s to almost 70 years in 2010–2014 [57].

## Women's participation in agricultural and non-agricultural activities

The impact of a job in the "agricultural/non-agricultural" sector on age at marriage was not significant. In India, agriculture continues to absorb and employ 2/3rd of the female workforce [58]. Meanwhile, the decline in women's share in the agricultural sector has been observed for decades [59]. However, in such states as Himachal Pradesh and Nagaland over 82% of women are still employed in the agricultural sector as agriculture is still the main livelihood of Scheduled Castes. Similar statistics are noted for Madhya Pradesh and Uttar Pradesh [58]. Many women also work as agricultural and daily laborers [40]. In the 1970s social changes led to an increase in employment opportunities in the non-agricultural sector, which reduced poverty and increased per capita income in all segments of Indian society. Unfortunately, in Scheduled Castes, this has progressed very slowly [60, 61]. Since the 1980s, attention has been focussed on multi-faceted programs encompassing, among others, education and employment of women [34]. These social and economic changes have resulted in the fact that relationships between caste, education and job opportunities have become more relaxed. Representatives of the lowest social group nowadays practice occupations previously unrelated to their caste, such as government jobs, teaching, retail, services, etc. [37]. As a consequence, a positive and significant correlation between women in the workforce and literacy rates has been observed [62].

## Conclusions

Over the last 40 years, age at marriage of women belonging to Scheduled Castes from the states of Madhya Pradesh and Uttar Pradesh has increased from 15 years on average to 18+ years (from girls to young women). This change has been influenced by a number of factors, but the main ones involve the development of women's education. A marked increase since the 1980s in the number of girls enrolled in schools in general, including girls belonging to Scheduled Castes, has led to a simultaneous decline of poverty and the number of child marriages. The implementation of laws prohibiting child marriages and regulating the age at marriage have also played a very important role in this process.

## Author Contributions

**Conceptualization:** Grażyna Liczbińska.

**Data curation:** Rajesh K. Gautam, Jyoti Jhariya, Arun Kumar.

**Formal analysis:** Marek Brabec.

**Funding acquisition:** Grażyna Liczbińska, Marek Brabec, Rajesh K. Gautam, Jyoti Jhariya, Arun Kumar.

**Investigation:** Rajesh K. Gautam, Jyoti Jhariya, Arun Kumar.

**Methodology:** Marek Brabec.

**Project administration:** Rajesh K. Gautam.

**Resources:** Grażyna Liczbińska, Marek Brabec, Rajesh K. Gautam, Jyoti Jhariya, Arun Kumar.

**Software:** Marek Brabec.

**Supervision:** Marek Brabec.

**Validation:** Marek Brabec.

**Visualization:** Grażyna Liczbińska, Marek Brabec.

**Writing – original draft:** Grażyna Liczbińska, Marek Brabec, Rajesh K. Gautam.

**Writing – review & editing:** Grażyna Liczbińska, Marek Brabec, Rajesh K. Gautam.

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
