## [Decision Letter · Decision Letter 0]

28 Oct 2022

PONE-D-22-19685From little girls to adult women: Changes in age at marriage in Scheduled Castes from Madhya Pradesh and Uttar Pradesh, IndiaPLOS ONE

Dear Dr. Liczbińska,

Thank you for submitting your manuscript to PLOS ONE. After careful consideration, we feel that it has merit but does not fully meet PLOS ONE’s publication criteria as it currently stands. Therefore, we invite you to submit a revised version of the manuscript that addresses the points raised during the review process.

We look forward to receiving your revised manuscript.

Kind regards,

Karina Cardoso Meira, Ph.D

Academic Editor

PLOS ONE

Journal Requirements:

“RKG and GL are grateful to the Ministry of Human Resources of the Government of India, Delhi for financial support (No. 174046N04) under the Global Initiative of Academic Network (GIAN). JJ expresses her gratitude to University Grants Commission, New Delhi, for providing financial assistance for pursuing her Ph.D. and awarding a Rajiv Gandhi National Fellowship (No.F.16–29/2006 (SA–II)/817) and Post-doctoral Fellowship for Women (No. F.No.15–2/2012). MB was partially supported by the long-term strategic development financing of the Institute of Computer Science (Czech Republic RVO 67985807).

5. We note that Figure 1 your submission contain [map/satellite] images which may be copyrighted. All PLOS content is published under the Creative Commons Attribution License (CC BY 4.0), which means that the manuscript, images, and Supporting Information files will be freely available online, and any third party is permitted to access, download, copy, distribute, and use these materials in any way, even commercially, with proper attribution. For these reasons, we cannot publish previously copyrighted maps or satellite images created using proprietary data, such as Google software (Google Maps, Street View, and Earth). For more information, see our copyright guidelines: http://journals.plos.org/plosone/s/licenses-and-copyright.

Additional Editor Comments (if provided):

The article is very interesting and reinforces the importance of the influence of education on social and economic outcomes related to girls and women. A topic of great relevance in the global agenda to reduce gender inequality. To improve the quality of the manuscript, it is necessary to carry out the corrections suggested by the reviewers.

Reviewers' comments:

Reviewer's Responses to Questions

**Comments to the Author**

1. Is the manuscript technically sound, and do the data support the conclusions?

Reviewer #1: Yes

Reviewer #2: Partly

2. Has the statistical analysis been performed appropriately and rigorously? 

Reviewer #1: Yes

Reviewer #2: Yes

3. Have the authors made all data underlying the findings in their manuscript fully available?

Reviewer #1: Yes

Reviewer #2: No

4. Is the manuscript presented in an intelligible fashion and written in standard English?

Reviewer #1: Yes

Reviewer #2: Yes

5. Review Comments to the Author

Reviewer #1: The article is very interesting and reinforces the importance of the influence of education on social and economic outcomes related to girls and women.

The introduction, method, results and discussion are adequately described, however I have some suggestions:

Materials and methods

a) Describe in more detail the genealogical method of age estimation, used to validate the information obtained;

b) Detail the “crosscheck” described on page 6, line 140

Results:

a) I suggest that the legends of the figures are next to them (Fig. 2, Fig. 3, Fig. 4);

b) Review the description in Table 2, as the data in the text is different from the data presented in the Table (Line 229-231, page 10).

Reviewer #2: The theme presented is current and relevant. Girls' vulnerabilities and penalties for maternity and marriage are exacerbated in contexts of poverty and social inequality. The article must be published, but with an adjustment in relation to the presentation of the database. The text is very well written and brings important results on gender inequalities in the private sphere. The work innovated by resorting to cohort analysis. This work complements the analyzes done for both India and looks at the positive impacts of education as a protective factor for child marriage.

The work presented is relevant to the literature and brings consistent results. Although the authors have made an effort to choose and apply the data analysis model, as well as to present and discuss the results, the same effort is not observed in the description of the database.

It would be interesting to specify more about the data source used. What are the criteria for choosing respondents and how can these criteria bias the sample? How does this sample resemble the female population of the studied locations? To what extent can the survey results, based on this sample, be generalized to the female population of the two states?

I could not identify, from the information provided, the place where the database could be consulted.

Small suggestions:

i) It would be interesting to indicate N in the regression model tables.

ii) The data presented in lines 293 to 309 would be easier to understand if they were presented in tables.

6. PLOS authors have the option to publish the peer review history of their article (what does this mean?). If published, this will include your full peer review and any attached files.

Reviewer #1: **Yes: **Juliano dos Santos

Reviewer #2: No

---

## [Author Response · Author response to Decision Letter 0]

25 Nov 2022

From little girls to adult women: Changes in age at marriage in Scheduled Castes from Madhya Pradesh and Uttar Pradesh, India

PLOS ONE.

Journal Requirements:

Dear Editor,

We are thankful for prompt processing the manuscript PONE-D-22-19685 entitled “From little girls to adult women: Changes in age at marriage in Scheduled Castes from Madhya Pradesh and Uttar Pradesh, India”. Here is point wise response to the comments: 

Response: 

We have made changes in the Manuscript as per suggested PLOS ONE style templates. 

Response: 

As suggested, the additional details regarding participant consent were provided under a subheading ‘Statement of Ethics’ in the Methods. Please refer to the main body of the text (pp. 8-9; lines: 178-192).

“RKG and GL are grateful to the Ministry of Human Resources of the Government of India, Delhi for financial support (No. 174046N04) under the Global Initiative of Academic Network (GIAN). JJ expresses her gratitude to University Grants Commission, New Delhi, for providing financial assistance for pursuing her Ph.D. and awarding a Rajiv Gandhi National Fellowship (No.F.16–29/2006 (SA–II)/817) and Post-doctoral Fellowship for Women (No. F.No.15–2/2012). MB was partially supported by the long-term strategic development financing of the Institute of Computer Science (Czech Republic RVO 67985807).The funders had no role in study design, data collection and analysis, decision to publish, or preparation of the manuscript.”

Response: 

As suggested, funding statement is revised please refer to page 19 (Lines 425-434).

Response: 

As suggested, a sub heading ‘Statement of Ethics’ in the Methods. Please refer to the main body of the text (pp. 8-9; lines: 178-192).

5. We note that Figure 1 your submission contain [map/satellite] images which may be copyrighted. All PLOS content is published under the Creative Commons Attribution License (CC BY 4.0), which means that the manuscript, images, and Supporting Information files will be freely available online, and any third party is permitted to access, download, copy, distribute, and use these materials in any way, even commercially, with proper attribution. For these reasons, we cannot publish previously copyrighted maps or satellite images created using proprietary data, such as Google software (Google Maps, Street View, and Earth). For more information, see our copyright guidelines: http://journals.plos.org/plosone/s/licenses-and-copyright.

Natural Earth (public domain): http://www.naturalearthdata.com/.

Response: 

As suggested, the Figure 1 is deleted. Accordingly, the Figure number also changed in the manuscript.

Response: 

As suggested, the references are updated. 

Additional Editor Comments (if provided):

The article is very interesting and reinforces the importance of the influence of education on social and economic outcomes related to girls and women. A topic of great relevance in the global agenda to reduce gender inequality. To improve the quality of the manuscript, it is necessary to carry out the corrections suggested by the reviewers.

Response: 

We are highly grateful for high words of appreciation. It will really open a new path for us to work more sincerely and contribute for the betterment of humanity. 

Reviewers' comments:

Reviewer's Responses to Questions

Comments to the Author

1. Is the manuscript technically sound, and do the data support the conclusions?

Reviewer #1: Yes

Reviewer #2: Partly

Response: 

We are thankful for positive comments and appreciation. The comments of second reviewer are also addressed and further details of data and ethical issues are provided in the revised version of manuscript. Please refer to page 8-9; lines: 178-192.________________________________________

2. Has the statistical analysis been performed appropriately and rigorously? 

Reviewer #1: Yes

Reviewer #2: Yes

Response: 

We are thankful for positive comments and appreciation. ________________________________________

3. Have the authors made all data underlying the findings in their manuscript fully available?

Reviewer #1: Yes

Reviewer #2: No

Response: 

Data Availability Statement is added in the revised version of the Manuscript. Please refer to Page 20, Line 435-436.

4. Is the manuscript presented in an intelligible fashion and written in standard English?

Reviewer #1: Yes

Reviewer #2: Yes

Response: 

Thanks to both reviewers. ________________________________________

5. Review Comments to the Author

Reviewer #1: 

The article is very interesting and reinforces the importance of the influence of education on social and economic outcomes related to girls and women.

Response: 

We are thankful for positive comments and appreciation.

The introduction, method, results and discussion are adequately described, however I have some suggestions:

Materials and methods

a) Describe in more detail the genealogical method of age estimation, used to validate the information obtained;

Response: 

In the revised manuscript detail the genealogical method of age estimation provided. Please refer to the main body of the text pages 6-7; lines:135-152.

b) Detail the “crosscheck” described on page 6, line 140

now line 149 – I have marked with green font!

Response: .

In the revised manuscript detail of “crosscheck” is further illustrated please refer to Page 7, Line 161-165. 

Results:

a) I suggest that the legends of the figures are next to them (Fig. 2, Fig. 3, Fig. 4).

Response: 

As suggested by the Reviewer, we revised the figures and now legends are next to them.

b) Review the description in Table 2, as the data in the text is different from the data presented in the Table (Line 229-231, page 10).

Response: 

In the revised Manuscript it is taken care that description of Table 2 and text is similar now. Please refer to page 12, Line 267-270. 

Reviewer #2: 

The theme presented is current and relevant. Girls' vulnerabilities and penalties for maternity and marriage are exacerbated in contexts of poverty and social inequality. The article must be published, but with an adjustment in relation to the presentation of the database. The text is very well written and brings important results on gender inequalities in the private sphere. The work innovated by resorting to cohort analysis. This work complements the analyses done for both India and looks at the positive impacts of education as a protective factor for child marriage.

Response: 

We are thankful for words of appreciation; it is really inspiring for us.

The work presented is relevant to the literature and brings consistent results. Although the authors have made an effort to choose and apply the data analysis model, as well as to present and discuss the results, the same effort is not observed in the description of the database.

Response: 

In the revised Manuscript, we tried to address this suggestion of the Reviewer.

It would be interesting to specify more about the data source used. What are the criteria for choosing respondents and how can these criteria bias the sample? How does this sample resemble the female population of the studied locations? To what extent can the survey results, based on this sample, be generalized to the female population of the two states?

Response: 

The Manuscript is revised and a paragraph is added to describe about the data source used, criteria for choosing respondents, how does this sample resemble the female population of the studied locations, etc. Please refer to page 6 Line 119-134. 

I could not identify, from the information provided, the place where the database could be consulted.

Response: 

In the revised Manuscript, the district from where the data was collected is mentioned please refer to Page 6 Line 132-134. 

Small suggestions:

i) It would be interesting to indicate N in the regression model tables.

Response: 

As suggested, in the revised manuscript the sample size (N) is indicated in the Tables (see: Table 1; p. 11 and Table 2; p. 12.

ii) The data presented in lines 293 to 309 would be easier to understand if they were presented in tables.

Response: 

As suggested, in the revised Manuscript data has been presented in Tables (see additional Table 3, p. 15 and Table 4, p.16.

---

## [Decision Letter · Decision Letter 1]

25 Jan 2023

From little girls to adult women: Changes in age at marriage in Scheduled Castes from Madhya Pradesh and Uttar Pradesh, India

PONE-D-22-19685R1

Dear Dr. Liczbińska

We’re pleased to inform you that your manuscript has been judged scientifically suitable for publication and will be formally accepted for publication once it meets all outstanding technical requirements.

Kind regards,

Karina Cardoso Meira, Ph.D

Academic Editor

PLOS ONE

Additional Editor Comments (optional):

Reviewers' comments:

Reviewer's Responses to Questions

**Comments to the Author**

1. If the authors have adequately addressed your comments raised in a previous round of review and you feel that this manuscript is now acceptable for publication, you may indicate that here to bypass the “Comments to the Author” section, enter your conflict of interest statement in the “Confidential to Editor” section, and submit your "Accept" recommendation.

Reviewer #1: All comments have been addressed

Reviewer #2: All comments have been addressed

2. Is the manuscript technically sound, and do the data support the conclusions?

Reviewer #1: Yes

Reviewer #2: Yes

3. Has the statistical analysis been performed appropriately and rigorously? 

Reviewer #1: Yes

Reviewer #2: Yes

4. Have the authors made all data underlying the findings in their manuscript fully available?

Reviewer #1: Yes

Reviewer #2: Yes

5. Is the manuscript presented in an intelligible fashion and written in standard English?

Reviewer #1: Yes

Reviewer #2: Yes

6. Review Comments to the Author

Reviewer #1: All suggested changes were accepted by the authors and the article.

The topic is relevant and deserves to be highlighted

Reviewer #2: (No Response)

7. PLOS authors have the option to publish the peer review history of their article (what does this mean?). If published, this will include your full peer review and any attached files.

Reviewer #1: **Yes: **Juliano dos Santos

Reviewer #2: No

---

## [Editor Report · Acceptance letter]

3 Feb 2023

PONE-D-22-19685R1 

From little girls to adult women: Changes in age at marriage in Scheduled Castes from Madhya Pradesh and Uttar Pradesh, India 

Dear Dr. Liczbińska:

I'm pleased to inform you that your manuscript has been deemed suitable for publication in PLOS ONE. Congratulations! Your manuscript is now with our production department. 

Kind regards, 

on behalf of

Dr. Karina Cardoso Meira 

Academic Editor

PLOS ONE